# Probiotic *Bifidobacterium breve* MCC1274 Mitigates Alzheimer’s Disease-Related Pathologies in Wild-Type Mice

**DOI:** 10.3390/nu14122543

**Published:** 2022-06-19

**Authors:** Mona Abdelhamid, Chunyu Zhou, Cha-Gyun Jung, Makoto Michikawa

**Affiliations:** Department of Biochemistry, Graduate School of Medical Sciences, Nagoya City University, 1 Kawasumi, Mizuho-cho, Mizuho-ku, Aichi, Nagoya 467-8601, Japan; monaahmed92@vet.bsu.edu.eg (M.A.); haruu5916@gmail.com (C.Z.)

**Keywords:** *Bifidobacterium breve* MCC1274, Aβ42 production, presenilin 1, tau phosphorylation, Akt/GSK-3β, glial activation, synapses

## Abstract

Probiotics improve brain function, including memory and cognition, via the microbiome–gut–brain axis. Oral administration of *Bifidobacterium breve* MCC1274 (*B. breve* MCC1274) improves cognitive function in *App^NL-G-F^* mice and mild cognitive impairment (MCI) subjects, and mitigates Alzheimer’s disease (AD)-like pathologies. However, its effects on wild-type (WT) mice have not yet been explored. Thus, the effects of *B. breve* MCC1274 on AD-like pathologies in two-month-old WT mice were investigated, which were orally administered *B. breve* MCC1274 for four months. Aβ levels, amyloid precursor protein (APP), APP processing enzymes, phosphorylated tau, synaptic protein levels, glial activity, and cell proliferation in the subgranular zone of the dentate gyrus were evaluated. Data analysis was performed using Student’s *t*-test, and normality was tested using the Shapiro–Wilk test. Oral administration of *B. breve* MCC1274 in WT mice decreased soluble hippocampal Aβ42 levels by reducing presenilin1 protein levels, and reduced phosphorylated tau levels. It also activated the protein kinase B (Akt)/glycogen synthase kinase-3β (GSK-3β) pathway, which may be responsible for the reduction in presenilin1 levels and inhibition of tau phosphorylation. *B. breve* MCC1274 supplementation attenuated microglial activation and elevated synaptic protein levels in the hippocampus. These findings suggest that *B. breve* MCC1274 may mitigate AD-like pathologies in WT mice by decreasing Aβ42 levels, inhibiting tau phosphorylation, attenuating neuroinflammation, and improving synaptic protein levels.

## 1. Introduction

Alzheimer’s disease (AD) is a neurodegenerative disease caused by a combination of genetic and environmental factors, and a lifestyle that gradually affects brain function. The brains of patients with AD reveal the presence of extracellular amyloid-β peptide (Aβ) deposition, forming senile plaques and hyperphosphorylated tau protein to form neurofibrillary tangles, which leads to neuronal cell death that causes cognitive decline and memory impairment [1]. The enzymatic cleavage of amyloid-β precursor protein (APP) by β- and γ-secretase produces Aβ [2]. Alternatively, APP can be cleaved by α-secretase to prevent Aβ generation. Neuroinflammation is also a common hallmark of AD, and is associated with Aβ and tau pathologies [3,4]. Activated microglia and astrocytes are involved in neuroinflammation in the AD brain, where their activation produces pro-inflammatory cytokines, such as tumor necrosis factor-alpha (TNF-α), interleukin-6 (IL-6), and IL-1β, subsequently resulting in neuronal cell death [5]. The most common sign of the disease is a decline in cognitive function and motor abilities, which ultimately leads to death [6]. AD cases are increasing worldwide, and the number of cases doubles every 20 years. The neurological pathologies of AD, including Aβ accumulation, tau phosphorylation, and neuroinflammation, progress slowly over more than 20 years before the onset of AD. Therefore, there is an urgent need to identify beneficial strategies for this disease before its onset.

Diet composition improvement represents a key factor to enhance animal health status and welfare [7,8]. Recently, it has become widely known that the daily intake of probiotics has a beneficial effect on human health; *Bifidobacterium* has immunomodulatory properties [9,10,11], and commensal *Bifidobacterium* nearly abolishes tumor outgrowth [12]. Several probiotic species have been reported to improve the intestinal barrier and maintain a healthy intestinal environment [13,14,15,16]. Moreover, some probiotics, such as *Lactobacillus* and *Bifidobacterium*, have been used as anti-aging agents [17]. Growing evidence suggests that supplementation with probiotics has the potential to prevent neurodegenerative diseases, including AD and Parkinson’s disease [18,19]. It is well known that probiotics influence brain function by one of these three pathways: endocrine pathway, immune modulation, or neural regulation [19]. Moreover, the administration of bifidobacteria improves brain function; *Bifidobacterium bifidum* BGN4, *Bifidobacterium breve*, and *Bifidobacterium Lactis* Probio-M8 have beneficial effects on cognition improvement, reduction of amyloidosis, and enhancement of synaptic function [20,21,22]. It has also been reported that some probiotics reduce phosphorylated tau levels by enhancing the PI3K/Akt/GSK-3β signaling pathway in the brains of AD-like model mice [23].

Previous studies demonstrated that oral supplementation with *Bifidobacterium breve* MCC1274 (*B. breve* MCC1274) improved cognitive function in both an AD-like mouse model and mild cognitive impairment (MCI) subjects [24,25]. Furthermore, recently, it has been reported that oral supplementation with *B. breve* MCC1274 for four months mitigated AD-like pathologies in *App^NL-G-F^* mice overexpressing humanized Aβ with APP mutations (Swedish (NL), Beyreuther/Iberian (F), and Arctic (G)) without overexpressing APP [26], as indicated by a reduction in amyloidosis, attenuation of microglial activation, and improvement of synaptic plasticity in the hippocampus, which ultimately prevented memory impairment [27]. These studies highlight the therapeutic potential of *B. breve* MCC1274 in preventing memory impairment and delaying the progression of AD. However, its effects on wild-type (WT) mice have not yet been explored. The current study aims to determine whether *B. breve* MCC1274 exerts a beneficial preventive effect on AD-related pathologies in WT mice. Two-month-old C57BL/6J mice orally supplemented with *B. breve* MCC1274 for four months mitigated AD-like pathologies by decreasing soluble Aβ42 levels, inhibiting tau phosphorylation, attenuating neuroinflammation, and improving synaptic protein levels in the hippocampus.

## 2. Materials and Methods

### 2.1. Preparation of Probiotic

*B. breve* MCC1274 was prepared as previously described [27]. Briefly, *B. breve* MCC1274 was isolated from infant feces. It was collected by centrifugation after growing it on a medium rich in yeast and glucose, lyophilized, and stored at 4 °C until use. Before oral administration to the experimental mice, *B. breve* MCC1274 were suspended in saline at a concentration of 1 × 10^9^ cfu/6.25 mg/200 μL.

### 2.2. Animals and Probiotic Supplementation

Two-month-old C57BL/6J mice were obtained from Japan SLC, Inc. (Hamamatsu, Japan) and housed under controlled humidity and temperature conditions with a 12-h light–dark cycle and *ad libitum* access to water and pellet food (Oriental Yeast Co., Tokyo, Japan). After one week of acclimatization, 40 mice were randomly assigned into 2 groups; one group (*n* = 20) received saline and the other group (*n* = 20) was administered 1 × 10^9^ cfu/6.25 mg/200 μL saline/mouse/day of *B. breve* MCC1274 via oral gavage five times/week for four months. At the end of the supplementation period, all mice were sacrificed under anesthesia, and their brains were collected for biochemical analyses and immunostaining. The animal experiment schedule is shown in Figure 1. All studies and protocols were approved by the Nagoya City University Institutional Care and Use of Laboratory Animals Committee, and the experiments were performed in accordance with the National Institute of Health Guide for the Care and Use of Laboratory Animals.

### 2.3. Western Blot Analysis

Both cortical and hippocampal tissues were removed quickly, and carefully placed in liquid nitrogen and weighed. They were then homogenized in ten volumes of pre-cooled RIPA buffer (50 mM Tris-HCl, 150 mM NaCl, 1% Nonidet P-40, 0.5% sodium deoxycholate, and 0.1% SDS, pH 7.6) containing protease and phosphatase inhibitor cocktails (FUJIFILM Wako Pure Chemical Corporation, Osaka, Japan). Then, the homogenates were centrifuged at 12,000 rpm for 30 min at 4 °C before collecting the supernatants. Protein concentration was determined using the Pierce^TM^ BCA Protein Assay kit (Thermo Fisher Scientific, Rockford, IL, USA) according to the manufacturer’s instructions. The homogenates were mixed with 4× sample buffer (125 mM Tris-HCl pH 6.8, 4% SDS, 20% glycerol, 2% 2-mercaptoethanol, and 40% bromophenol blue), followed by heating for 7 min at 95 °C. Protein lysates (20 µg) were separated using SDS-PAGE and transferred to Immobilon-P membranes (Millipore, Billerica, MA, USA). These membranes were blocked with 5% skim milk in Tris-buffered saline (TBS) containing 0.1% Tween 20 (TBS-T) for 1 h at room temperature (RT), and incubated overnight at 4 °C with the following primary antibodies: anti-APP (1:1000, Millipore), anti-ADAM10 (1:1000, Millipore), anti-BACE1 (1:1000, R&D, Minneapolis, MN, USA), anti-PS1 (1:1000, Millipore), anti-β-amyloid (6E10, 1:1000, Biolegend, San Diego, CA, USA), anti-sAPPβ (1:1000, IBL, Gunma, Japan), anti-Synaptotagmin (SYP, 1:1000, BD Biosciences, San Jose, CA, USA), anti-Syntaxin (1:1000, Sigma-Aldrich, St. Louis, MO, USA), anti-Synaptophysin (SYP, 1:20,000, Abcam, Cambridge, UK), anti-PSD95 (1:1000, #3450, Cell Signaling, Danvers, MA, USA), anti-pAKT (S473, 1:500, Cell Signaling), anti-AKT (1:1000, Cell Signaling), anti-pGSK3α/β (S21/9, 1:500, Cell Signaling), anti-GSKα/β (1:1000, Cell Signaling), anti-tau (Tau5, 1:1000, Biolegend), anti- AT180 (T231, 1:500, Invitrogen, Carlsbad, CA, USA), anti-AT8 (S202/T205, 1:500, Thermo Fisher Scientific), anti-JNK (1:1000, Cell signaling), anti-pJNK (1:1000, Cell Signaling), anti-mTOR (1:1000, #2972, Cell Signaling), anti-pmTOR (S2448) (1:1000, Cell Signaling), anti-pmTOR (S2481) (1:1000, Cell Signaling), anti-p53 (1C12) (1:1000, Cell Signaling), anti-pp53 (Ser15, 1:1000, Cell Signaling), anti-Iba1 (1:1000, Wako), anti-GFAP (1:1000, Sigma-Aldrich), and anti-actin (1:5000, Proteintech, Tokyo, Japan) antibodies. The membranes were washed with TBS-T and incubated with appropriate horseradish peroxidase-conjugated antibodies. Chemiluminescence signals were visualized using Immunostar Zeta or Immunostar LD (FUJIFILM Wako Pure Chemical Corporation), and analyzed using an Amersham Imager 680 (GE Healthcare, Marlborough, MA, USA). Actin was used as an internal control, and densitometric analysis was performed using the ImageJ software (NIH, Bethesda, MD, USA). The ratio of the signal for each target protein to that of actin is depicted as the fold change of the control.

### 2.4. Assessment of the Levels of Soluble Aβ in the Brain by ELISA

Both the cortex and hippocampus (*n* = 7 per group) were weighed and homogenized in 10× TBS buffer. Cortical and hippocampal homogenates were centrifuged at 100,000 rpm for 20 min at 4 °C. The supernatant was stored at −80 °C until use for detection of soluble Aβ40 and Aβ42 by using sandwich Aβ ELISA kits (Wako Pure Chemical Industries) by following the manufacturer’s instructions. In brief, the standard and samples were incubated overnight at 4 °C, followed by washing and incubation for 1 h at 4 °C with HRP-conjugated antibody solution, and then incubation for 30 min at RT with TMB solution after extensive washing. The stop solution was then added, and the absorbance of each well was read at 450 nm. The levels of Aβ are presented as pmol of Aβ per gram of protein. All samples were run in duplicate for each analyte, and exhibited parallel displacement to the standard curve.

### 2.5. Tissue Processing for Immunohistochemistry and BrdU Staining

At the end of the supplementation period, seven mice per group were administered three consecutive daily (twice per day) intraperitoneal injections of 5-Bromo-2-deoxy-uridine (BrdU) (Wako). Mice were anesthetized with sevoflurane, and intracardially perfused with phosphate buffer saline (PBS), and then 4% paraformaldehyde (PFA) solution. Brains were post-fixed in 4% PFA for 3 days at 4 °C after removal from the skull, and transferred to 30% sucrose for 2–3 days. Thereafter, the brains were frozen by immersion in dry ice, embedded in optimal cutting temperature compound, cut into 40 μm-thick sections on a vibratome (Leica Microsystems, Wetzlar, Germany), placed in a cryoprotection solution (30% sucrose, 150 mM NaCl, and 250 mM polyvinylpyrrolidone in 0.1 M PB), and stored at −20 °C until used for staining.

### 2.6. Immunofluorescence Staining and Cell Proliferation Analysis

Antigen retrieval for brain slides was achieved by boiling in citrate buffer for 5 min followed by cooling at RT. Brain sections were blocked with 5% goat serum/0.25% Triton X/PBS for 30 min. The sections were incubated overnight at 4 °C with the primary antibodies against Iba1 (1:200, WAKO) and GFAP (1:100, Sigma-Aldrich). After washing, the sections were incubated with goat anti-rabbit Alexa Fluor 568 or goat anti-mouse Alexa Fluor 488 secondary antibody (Thermo Fisher Scientific) for 1 h at RT in the dark. Finally, the cells were mounted with DAPI solution (Vector Laboratories Inc., Burlingame, CA, USA). Images were acquired with a confocal microscope (Olympus, Southall, UK). The number of Iba1- and GFAP-positive cells was quantified in randomly selected two fields (1 × 1 mm^2^) of the hippocampus and four fields (1 × 1 mm^2^) of the cortex per mouse using ImageJ. BrdU staining was performed using a BrdU Labeling and Detection kit (Roche, San Francisco CA, USA). Briefly, sections were passed through antigen retrieval, blocking solution, and then incubated with the anti-BrdU antibody for 45 min at 37 °C, followed by incubation with anti-mouse FITC antibody for 30 min at 37 °C. Finally, they were cover-slipped with DAPI solution. BrdU-positive cells were imaged using a confocal microscope (Olympus) and quantified in eight sections per mouse using ImageJ software.

### 2.7. Statistical Analysis

GraphPad Prism software (San Diego, CA, USA) was used for statistical analysis. All data are presented as mean ± SD. Results were analyzed using a two-tailed unpaired Student’s *t*-test (for normally distributed variables). Significance was defined when *p* < 0.05. Normality was tested using the Shapiro–Wilk test. The results of these tests indicate that the data for both saline and probiotic groups are normally distributed.

## 3. Results

### 3.1. B. breve MCC1274 Supplementation Reduces Soluble Aβ42 Level and Presenilin1 (PS1) Protein Level in the Hippocampus

Aβ production in the brain is affected by altered levels of APP and APP-processing enzymes, such as ADAM10, BACE1, and PS1, which are α-secretase, β-secretase, and γ-secretase components, respectively. A previous study demonstrated that oral supplementation with *B. breve* MCC1274 reduced hippocampal Aβ production in *App^NL-G-F^* mice by increasing the ADAM10 levels [27]. Thus, the effect of probiotic supplementation on Aβ levels in WT mice was investigated, and the results showed a significant decrease in soluble Aβ42 levels in the hippocampal extracts of mice supplemented with probiotics compared to those of mice that received saline (Figure 2A). However, soluble Aβ40 levels in the hippocampus (Figure 2A), as well as soluble Aβ40 and Aβ42 levels in the cortex, were not affected by probiotic supplementation (Figure 2B). Intra-assay coefficients of variation (CVs) were calculated by dividing the standard deviation by the duplicate mean and multiplying by 100. CVs for soluble Aβ40 and Aβ42 in both the cortex and hippocampus ranged from 0.57 to 6.9%, 1.6 to 7.1%, 0.6 to 7.7%, and 1.5 to 6.7%, respectively.

Next, the protein levels of APP and APP-processing enzymes were evaluated by Western blotting, and the results showed that mice supplemented with *B. breve* MCC1274 had considerably lower protein levels of PS1 in the hippocampus, but not in the cortex than those in mice that received saline (Figure 3). In contrast, the protein levels of APP, ADAM10, and BACE1 in both the cortex and hippocampus were not substantially different between the two groups (Figure 3). These findings indicate that *B. breve* MCC1274 supplementation reduced soluble Aβ42 levels in the hippocampus of WT mice by decreasing PS1 protein levels.

### 3.2. B. breve MCC1274 Supplementation Activates Akt/GSK-3β Pathway in the Hippocampus

Next, the underlying molecular mechanism by which probiotic supplementation decreased PS1 protein levels was investigated. It has been reported that an increase in Akt phosphorylation at S473 causes an increase in the level of phosphorylated (p)-GSK-3β at S9, which is an inactive form of GSK-3β that, in turn, contributes to the reduction in PS1 levels [28]. Thus, the protein levels of p-Akt (S473) and p-GSK-3β (S9) were assessed using Western blotting, and the results showed that p-Akt and p-GSK-3β protein levels were significantly increased in the hippocampus of the probiotic-supplemented group compared to those in the saline-supplemented group (Figure 4A). However, the protein levels of p-GSK-3α (S21) in the hippocampus (Figure 4A), as well as those of p-Akt and p-GSK-3α/β in the cortex, did not differ between the two groups (Figure 4B). It has also been reported that the inhibition of c-Jun N-terminal kinase (JNK) activity and overexpression of p53 reduce APP processing through suppression of PS1 levels [29,30], and that inhibition of the mammalian target of rapamycin (mTOR) phosphorylation suppresses PS1 levels [31]. Therefore, their protein levels were assessed by Western blotting. Unfortunately, no substantial differences were found in both total and phosphorylated JNK, p53, or mTOR levels between the two groups in both the cortex and hippocampus (Appendix A). Taken together, these findings indicate that oral supplementation of *B. breve* MCC1274 in WT mice reduces hippocampal PS1 protein levels through the Akt/GSK-3β-dependent pathway, leading to a reduction in Aβ42 generation.

### 3.3. B. breve MCC1274 Supplementation Inhibits Tau Hyperphosphorylation in the Hippocampus

Tau hyperphosphorylation is believed to be crucial in the pathogenesis of AD because it promotes neurofibrillary tangle formation in the brains of patients with AD, and causes neuronal cell death [32,33]. In addition, Akt activation inhibits GSK-3β activation, which is known to decrease tau phosphorylation at T231, S202, and T202 [34]. Therefore, the effect of *B. breve* MCC1274 supplementation on both total and p-tau (T231 and S202/T205) protein levels was investigated using Western blotting. The results showed lower levels of phosphorylated tau at T231, S202, and T205 in the hippocampus of the probiotic-supplemented group than in the saline-supplemented group, whereas total tau protein levels did not differ between the two groups in the hippocampus (Figure 5A). However, there was no change in total and phosphorylated tau protein levels in the cortex (Figure 5B). These results indicate that oral supplementation with *B. breve* MCC1274 inhibits hippocampal tau hyperphosphorylation in WT mice through the Akt/GSK-3β-dependent pathway.

### 3.4. B. breve MCC1274 Supplementation Enhances Synaptic Plasticity in the Hippocampus

Because of the essential role of synaptic proteins in the early development of neurons and regulation of neurotransmitter release, the protein levels of the scaffolding postsynaptic protein, PSD-95, and three different presynaptic proteins, synaptotagmin (SYT), syntaxin, and synaptophysin (SYP), in both the hippocampus and cortex of WT mice, were assessed using Western blotting. Supplementation with *B. breve* MCC1274 significantly increased the protein levels of SYT and syntaxin, and showed a tendency to increase the protein levels of SYP and PSD-95 in hippocampal extracts (Figure 6A). In contrast to the hippocampus, these protein levels did not change in the cortical extracts between the two groups (Figure 6B). These findings indicate that *B. breve* MCC1274 supplementation increases the expression of synaptic proteins in WT mice, which is consistent with previous findings that *B. breve* MCC1274 supplementation increases these synaptic protein levels in *App^NL-G-F^* mice [27].

### 3.5. B. breve MCC1274 Supplementation Attenuated Microglial Activation in the Hippocampus

As activated microglia and astrocytes are involved in AD-like pathologies, we also investigated whether *B. breve* MCC1274 supplementation alters their levels in the brains of WT mice. Brain sections were stained with anti-Iba1 (a marker for activated microglia) and anti-GFAP (a marker for reactive astrocytes) antibodies, and the numbers of Iba1^+^ and GFAP^+^ cells were quantified. In the hippocampus of mice, supplementation with *B. breve* MCC1274 remarkably decreased the number of Iba1^+^ cells compared with those in vehicle treatment (Figure 7A). However, the number of GFAP^+^ cells in the hippocampus showed no difference between the two groups (Figure 8A). Although both Iba1^+^ and GFAP^+^ cells were rarely detected in the cortex of WT mice, Iba1^+^ and GFAP^+^ cells around the brain surface were observed. No differences were found in the number of Iba1^+^ cells in the cortex (Figure 7B) or GFAP^+^ cells in the hippocampus and cortex (Figure 8) between the two groups. Furthermore, the protein levels of Iba1 and GFAP were investigated using Western blotting, and obtained similar results to those of immunostaining analysis. That is, the Iba1 protein level was reduced in the hippocampus of the probiotic-supplemented group compared to that in the saline-supplemented group, whereas Iba1 protein level in the cortex, as well as GFAP protein levels in both the cortex and hippocampus, were not substantially different between the two groups (Figure 9A,B). As microglial activation is associated with the production of pro-inflammatory cytokines [35,36], the mRNA levels of pro-inflammatory cytokines (TNF-α, IL-6, and IL-1β) in both hippocampal and cortical extracts were assessed using qRT-PCR analysis. Unfortunately, their mRNA levels were undetectable, even in the saline-supplemented group (data not shown), possibly because neuroinflammation had not occurred because the mice were still young. These results suggest that *B. breve* MCC1274 supplementation in WT mice attenuates microglial activation only in the hippocampus, and does not affect reactive astrocytes, which is consistent with previous findings in *App^NL-G-F^* mice [27].

### 3.6. B. breve MCC1274 Supplementation Does Not Alter Cell Proliferation in the Hippocampus

To assess cell proliferation in the subgranular zone of the dentate gyrus after oral supplementation with *B. breve* MCC1274, brain sections were labeled with BrdU. The number of BrdU^+^ cells did not change in the probiotic-supplemented group compared to that in the saline-supplemented group (Figure 9C). These results suggest that oral supplementation with *B. breve* MCC1274 does not alter cell proliferation in the hippocampus of WT mice, which is consistent with the previous finding in *App^NL-G-F^* mice [27].

## 4. Discussion

In recent decades, several clinical and experimental studies have explored the role of microbiota in modulating the gut–brain axis because the enteric microbiota plays a vital role in the gut–brain axis not only through the intrinsic nervous system, but also through hormonal and metabolic pathways [37,38,39,40,41]. A previous study demonstrated that oral supplementation with *B. breve* MCC1274 suppressed hippocampal neuroinflammation in Aβ-injected mice, which led to the amelioration of cognitive dysfunction [24]. Furthermore, MCI subjects who received this probiotic for 16 weeks showed improved memory function [25]. In addition, recently, it has been found that AD-like pathologies in *App^NL-G-F^* mice supplemented with *B. breve* MCC1274 were mitigated through enhancement of the non-amyloidogenic pathway, as well as attenuation of neuroinflammation, which finally prevented memory deficits [27]. However, the effects of this probiotic on AD-like pathologies in WT mice have remained unclear. In this study, the effect of *B. breve* MCC1274 on AD-like pathologies in WT mice was explored. A similar beneficial effect was also exerted in these mice by orally-supplemented *B. breve* MCC1274, as demonstrated by a reduction in soluble Aβ42 levels, attenuation of neuroinflammation, and improvement of synaptic plasticity in the hippocampus.

Aggregation of Aβ in the brain is the most remarkable characteristic of AD, which is generated through the cleavage of APP by β-secretase and then by γ-secretase. PS1 is a key enzyme in the generation of Aβ because it harbors the catalytic site of the γ-secretase complex [42]. Furthermore, an in vivo study demonstrated that the overexpression of PS1 is sufficient to elevate the production and aggregation of Aβ in the brain, and mutant PS1 produces greater deposition of Aβ in the brain, as it cleaves APP more efficiently at the Aβ42 site than at the Aβ40 site [43]. These findings showed that *B. breve* MCC1274 supplementation was able to downregulate hippocampal soluble Aβ42 production in WT mice by decreasing the protein levels of the γ-secretase component of PS1. A previous study demonstrated that *B. breve* MCC1274 supplementation decreased Aβ production and deposition in *App^NL-G-F^* mice through upregulation of ADAM10 protein levels [27]. In the current study, *B. breve* MCC1274 supplementation altered the protein levels of PS1, but not ADAM10. This difference between WT and *App^NL-G-F^* mice suggests that there are different regulatory mechanisms underlying the *B. breve* MCC1274-reduced Aβ42 generation.

Brain dysfunction may be linked to the downregulation of Akt and elevation of the inactive form of GSK-3β [44]. Furthermore, Akt activation may be considered a potential preventive therapy for neurodegenerative diseases [45,46]. It has also been reported that Akt activation with an elevation of the inactive form of GSK-3β contributes to the downregulation of PS1 protein levels [28]. In this study, the results showed that *B. breve* MCC1274 supplementation increased pAkt (S473) and the inactive form of pGSK-3β (S9), without significant changes in pGSK-3α (S21) in the hippocampi of WT mice. According to a previous study, the inactivation of GSK-3β results from the increased phosphorylation at the S9 site [47]. Another study demonstrated that increasing the inactive form of pGSK-3β resulted in the attenuation of PS1/γ-secretase activity [48]. Taken together, these findings suggest that *B. breve* MCC1274 supplementation in WT mice reduces PS1 protein levels through the Akt/GSK-3β signaling pathway.

Recently, abnormally hyperphosphorylated tau has attracted considerable attention as a targeted drug for AD [49,50]. In this study, the results showed that oral supplementation with *B. breve* MCC1274 downregulated the levels of some tau phosphorylation sites (T231, S202, and T205) in the hippocampus of WT mice, which is different from the previous findings in *App^NL-G-F^* mice because tau phosphorylation could not be altered by *B. breve* MCC1274 supplementation in these mice [27]. In this study, the increased phosphorylation levels of both Akt and GSK-3β, without changes in the total protein levels of these two kinases, participate in the inhibition of tau hyperphosphorylation in the hippocampus. Therefore, *B. breve* MCC1274 supplementation appears to be a potential therapeutic approach for the prevention of tau hyperphosphorylation in WT mice.

It has been reported that impaired synaptic plasticity is found in several neurodegenerative diseases, including AD, and is also an early event in the disease [51]. Synaptic dysfunction is hypothesized to be not only the main hallmark, but also a primary cause of AD [52]. Indeed, synaptic dysfunction is caused by the aggregation of soluble Aβ oligomers that bind to both presynaptic and postsynaptic proteins and impair their function [53]. The levels of four synaptic proteins, SYT, syntaxin, SYP, and PSD95, were determined in both the cortex and hippocampus. The results indicated that supplementation with *B. breve* MCC1274 substantially increased these protein levels in the hippocampus of WT mice. Furthermore, previous findings demonstrated that this probiotic recovered both pre-and postsynaptic protein levels in *App^NL-G-F^* mice [27]. Thus, the current and previous findings suggest that *B. breve* MCC1274 may be used as a targeted therapy to prevent synaptic dysfunction by increasing both pre-and postsynaptic protein expression in both AD-like models and WT mice.

Neuroinflammation caused by activated astrocytes and microglia contributes to the progression of neurodegenerative diseases [3,4,54,55]. A previous study demonstrated that *B. breve* MCC1274 supplementation has an anti-inflammatory effect in the brains of *App^NL-G-F^* mice, which might be achieved by shifting microglia from the M1 to M2 phenotype, resulting in the elevation of anti-inflammatory cytokines and the reduction of pro-inflammatory cytokines [27]. In the present study, the possible anti-inflammatory effects of this probiotic were demonstrated in WT mice. The results of immunofluorescence staining and Western blotting indicated a substantial downregulation of Iba1^+^ cell number and protein level in the hippocampus of WT mice, whereas there was no change in GFAP^+^ cell number or protein levels. These results consistently suggest that *B. breve* MCC1274 has a neuroprotective effect in the hippocampus of WT mice through its anti-inflammatory effect, which may be achieved by reducing activated microglia. Several studies have shown that the dysregulation of the PI3K/Akt signaling pathway leads to microglial activation, which promotes the production of pro-inflammatory cytokines, such as IL-6 and TNF-α [56,57]. Therefore, enhanced Akt activation in the brains of mice supplemented with *B. breve* MCC1274 may play an important role in the attenuation of microglial activation. However, the underlying mechanism by which *B. breve* MCC1274 supplementation activates Akt is still unclear. Several studies have demonstrated that some probiotics elevate Akt phosphorylation [58,59]. However, further investigations are required to understand the mechanism of *B. breve* MCC1274-mediated activation of the Akt-GSK-3β pathway.

Interactions between microglia and synapses have been observed in most neurodegenerative diseases. It has been reported that increased microglial activation, along with a reduction in both the number and function of synapses, occurs in the early development of AD [60,61]. Activated microglia secrete pro-inflammatory cytokines that negatively affect synaptic transmission and activity, resulting in impaired cognitive function [62,63,64]. Therefore, this study suggests that the attenuation of microglial activation in the hippocampus of WT mice by *B. breve* MCC1274 supplementation might play an essential role in the improvement of synaptic protein expression.

As neurogenesis plays a vital role in improving cognitive function and mental health [65], one of the objectives of this study was to investigate the effect of *B. breve* MCC1274 supplementation on neurogenesis in the brains of WT mice. After four months of probiotic supplementation, no remarkable differences were observed in the number of BrdU^+^ cells in the SVZ of the DG area in these mice, which is consistent with previous findings in *App^NL-G-F^* mice [27]. Thus, these results suggest that *B. breve* MCC1274 supplementation does not affect hippocampal neurogenesis in WT or *App^NL-G-F^* mice.

## 5. Conclusions

Previous findings showed that oral administration of *B. breve* MCC1274 improved cognitive function and mitigated AD-like pathologies. However, its effect on WT mice is unknown. In this study, the results showed that *B. breve* MCC1274 supplementation collectively decreased soluble Aβ42 and PS1 protein levels through the Akt/GSK-3β signaling pathway, attenuated microglial activation, and inhibited tau phosphorylation, which might improve synaptic function. *B. breve* MCC1274 supplementation had similar potential therapeutic effects on both WT and *App^NL-G-F^* mice, except for the difference in the underlying regulatory mechanism of *B. breve* MCC1274 supplementation-reduced Aβ generation and the impact on tau hyperphosphorylation. Overall, this study and the previous studies support the idea that oral supplementation with *B. breve* MCC1274 has a beneficial effect on multiple AD therapeutic targets.

## Figures and Tables

**Figure 1 nutrients-14-02543-f001:**
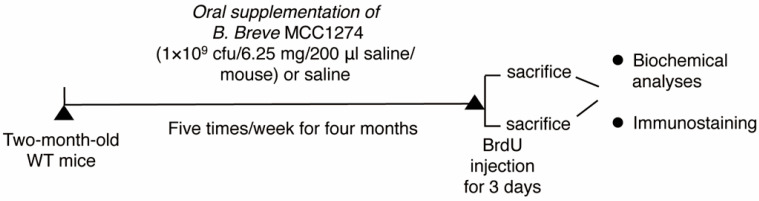
A schematic figure explaining the timeline of the experiment. Two-month-old wild-type (WT) mice were randomly divided into saline and probiotic groups: the saline group received saline, and the probiotic group (*n* = 20 per group) was supplemented with *B. breve* MCC12741 (1 × 10^9^ cfu/6.25 mg/200 μL saline/mouse) via oral gavage five times/week for 4 months. At the end of supplementation, mice were injected with BrdU and were then sacrificed, and their brains were collected for immunostaining and biochemical analyses.

**Figure 2 nutrients-14-02543-f002:**
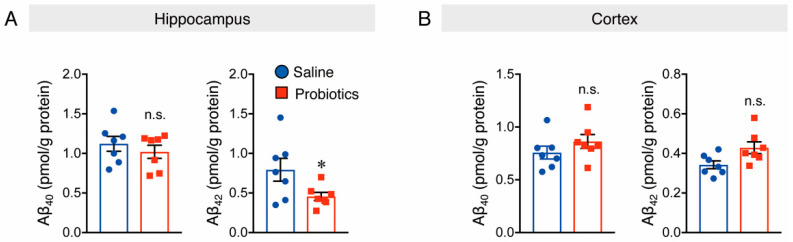
Oral administration of *B. breve* MCC1274 reduces soluble hippocampal Aβ42 levels in WT mice. Soluble hippocampal Aβ40 and Aβ42 levels (**A**) and soluble cortical Aβ40 and Aβ42 levels (**B**) in WT mice were measured by Sandwich ELISA. Aβ levels were normalized to each protein concentration. Data are represented as the mean ± SD, *n* = 7 per group, n.s. no significant difference, * *p* < 0.05 compared with the saline group, data analyzed by Student’s *t*-test.

**Figure 3 nutrients-14-02543-f003:**
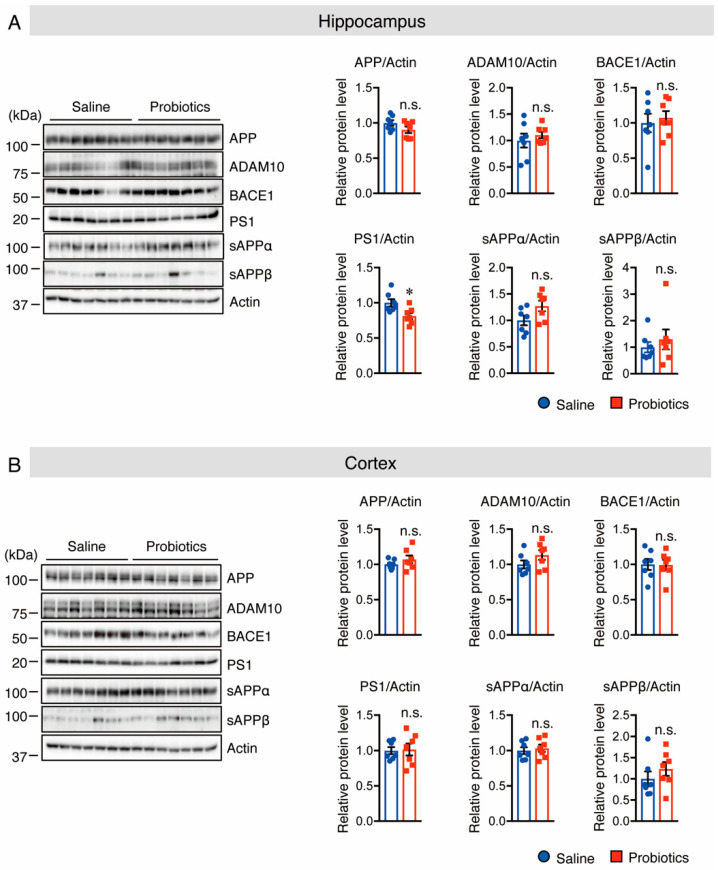
Oral supplementation of *B. breve* MCC1274 decreases PS1 protein level in the hippocampus of WT mice. Western blot analysis of APP, ADAM10, BACE1, PS1, sAPPα, sAPPβ, and actin in the hippocampus (**A**) and cortex (**B**). The protein levels were quantified by densitometry, normalized to the actin level, and expressed as a relative protein level. Data are represented as mean ± SD; *n* = 7 per group; n.s. no significant difference; * *p* < 0.05 compared with the saline group, as determined by Student’s *t*-test.

**Figure 4 nutrients-14-02543-f004:**
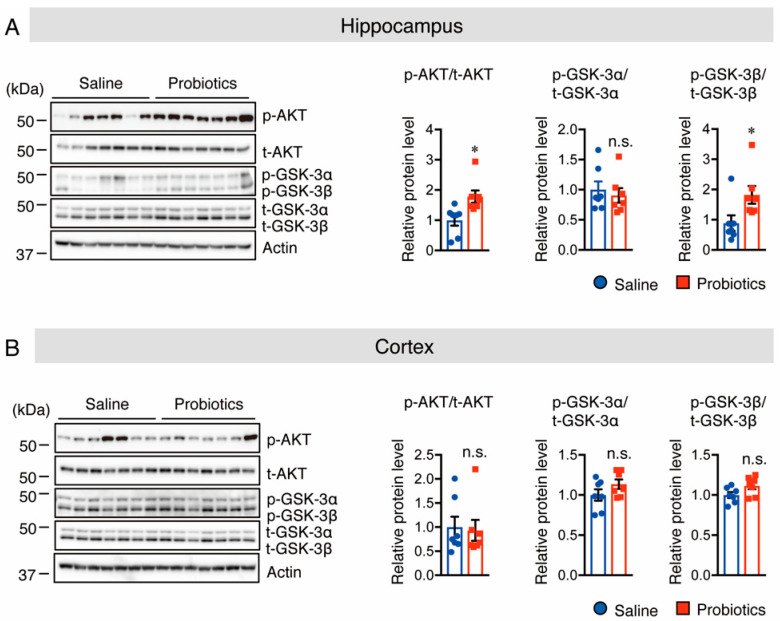
Oral supplementation of *B. breve* MCC1274 increases Akt and p-GSK-3β phosphorylation levels, but not GSK-3α phosphorylation levels, in the hippocampus of WT mice. Western blot analysis of phosphorylated (p-) Akt, total (t-) Akt, p-GSK-3α/β, t-GSK-3α/β, and actin in the hippocampus (**A**) and cortex (**B**). Quantification of phosphorylated (p-) protein levels normalized to total (t-) protein levels and expressed as values relative to the control. Data are represented as mean ± SD, *n* = 7 per group; n.s. no significant difference; * *p* < 0.05 compared with the saline group, as determined by Student’s *t*-test.

**Figure 5 nutrients-14-02543-f005:**
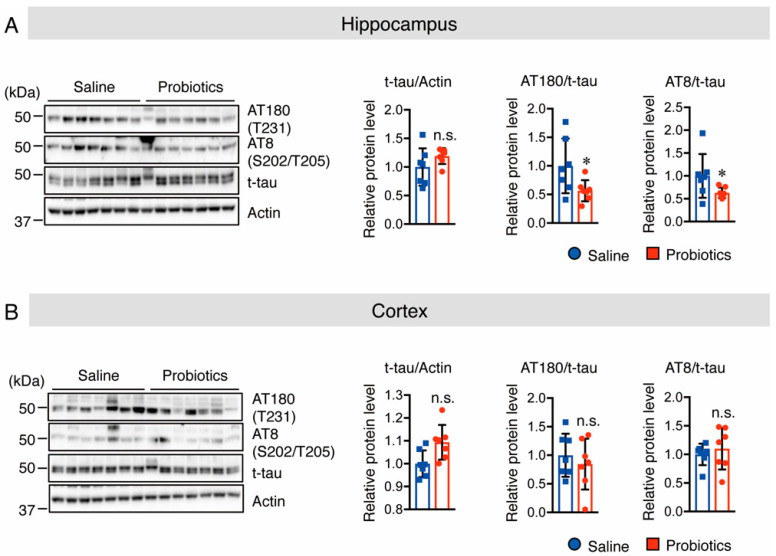
Oral supplementation of *B. breve* MCC1274 inhibits tau phosphorylation in the hippocampus of WT mice. Western blot analysis of tau phosphorylation at site T231 (AT180), S202/T205 (AT8), total tau, and actin in the hippocampus (**A**) and cortex (**B**). Quantification of phosphorylated (p-) protein levels normalized to total (t-) protein levels and expressed as values relative to the control. Data are represented as mean ± SD; *n* = 7 per group; n.s. no significant difference; * *p* < 0.05 compared with the saline group, as determined by Student’s *t*-test.

**Figure 6 nutrients-14-02543-f006:**
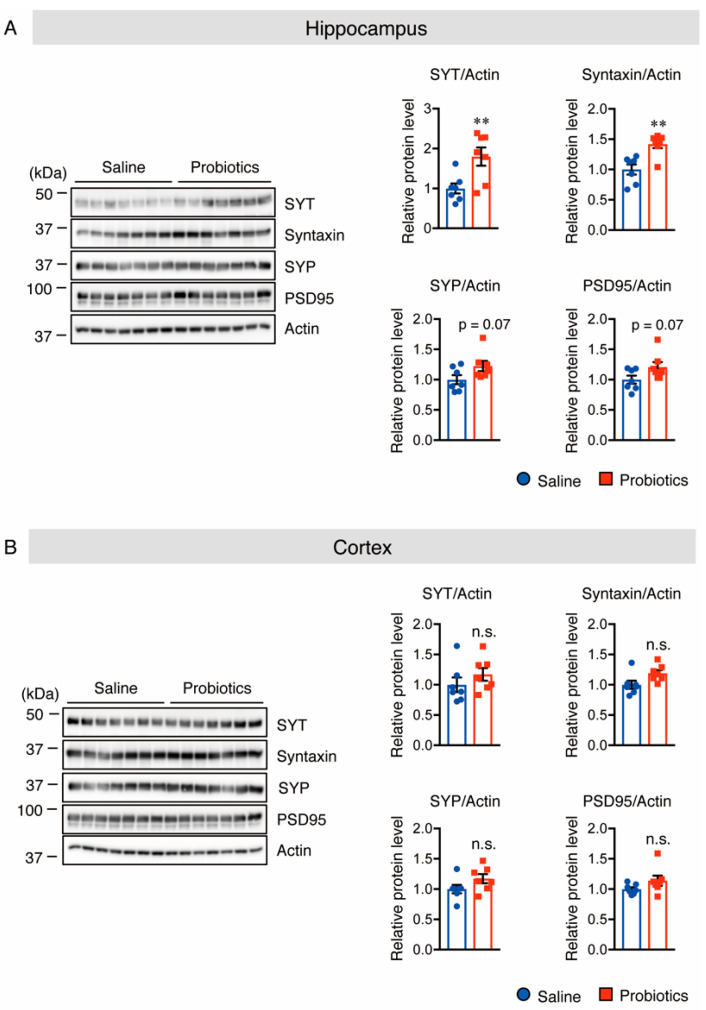
Oral supplementation of *B. breve* MCC1274 upregulates synaptic protein levels in the hippocampus of WT mice. Western blot analysis of SYT, syntaxin, SYP, PSD95, and actin in the hippocampus (**A**) and cortex (**B**). The protein levels were quantified by densitometry, normalized to the actin level, and expressed as a relative protein level. Data are represented as mean ± SD; *n* = 7 per group; n.s. no significant difference; ** *p* < 0.01 compared with the saline group, as determined by Student’s *t*-test.

**Figure 7 nutrients-14-02543-f007:**
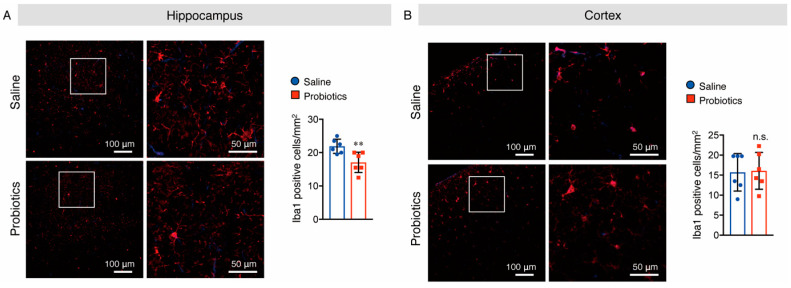
Oral supplementation of *B. breve* MCC1274 attenuates microglial activation in the hippocampus of WT mice. Iba1 immunostaining in the hippocampus (**A**) and cortex (**B**) of WT mice. Representative images of brain sections stained with an anti-Iba1 antibody (red) and DAPI nuclear counterstain (blue) (left panel). The number of Iba1^+^ cells in the hippocampus (**A**) and cortex (**B**) (right panel). Data are represented as mean ± SD, *n* = 6 per group; n.s. no significant difference; ** *p* < 0.01 compared with the saline group, as determined by Student’s *t*-test.

**Figure 8 nutrients-14-02543-f008:**
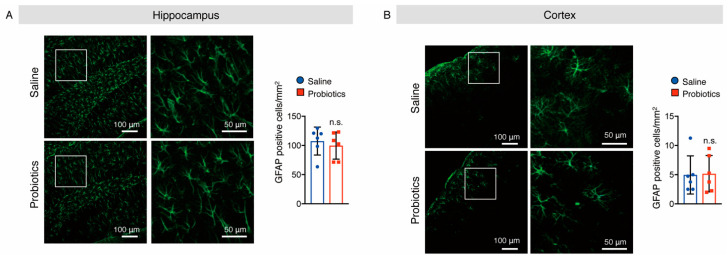
Oral supplementation of *B. breve* MCC1274 does not affect astrogliosis in the brain of WT mice. GFAP immunostaining in the hippocampus (**A**) and cortex (**B**) of WT mice. Representative images of brain sections stained with an anti-GFAP antibody (green) and DAPI nuclear counterstain (blue) (left panel). The number of GFAP^+^ cells in the hippocampus (**A**) and cortex (**B**) (right panel). Data are represented as mean ± SD, *n* = 6 per group; n.s. no significant difference compared with the saline group, as determined by Student’s *t*-test.

**Figure 9 nutrients-14-02543-f009:**
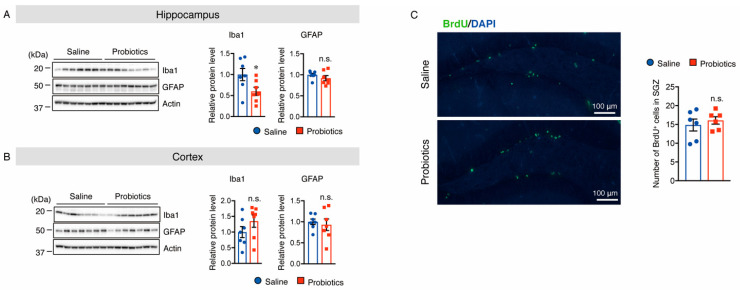
Oral supplementation of *B. breve* MCC1274 reduces the protein level of Iba1, but not GFAP, in the hippocampus of WT mice. Western blot analysis of Iba1, GFAP, and actin in the hippocampus (**A**) and cortex (**B**). The protein levels were quantified by densitometry, normalized to the actin level, and expressed as a relative protein level. (**C**) Oral supplementation of *B. breve* MCC1274 does not affect adult hippocampal neurogenesis in WT mice. Photomicrographs of brain sections throughout the hippocampal dentate gyrus (DG). Immunofluorescence staining with BrdU (green) and DAPI nuclear counterstain (blue) (left panel). BrdU (50 mg/kg, intraperitoneal injection) was injected twice/day for 3 days. The BrdU^+^ cell numbers in the SVZ of the DG area were quantified (right panel). Data are represented as mean ± SD, *n* = 7 per group; n.s. no significant difference; * *p* < 0.05 compared with the saline group, as determined by Student’s *t*-test.

## Data Availability

All data used in this study are available from the corresponding authors on reasonable request.

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
