# Peer review of "Probiotic Bifidobacterium breve MCC1274 Mitigates Alzheimer’s Disease-Related Pathologies in Wild-Type Mice"

_nutrients, 2022, doi:10.3390/nu14122543_

Round 1

Reviewer 1 Report

I recommended avoiding the use of personal forms (e.g. we, our etc.) throughout the text.

The title as well as keywords accurately reflects the major findings of the work.

The abstract adequately summarize methodology, results, and significance of the study. However, Authors should indicate the statistical analysis applied on data.

The introduction section falls within the topic of the study, however, Authors should enhance this section adding more information concerning the diet supplementation emphasizing the significant increase of interest showed by scientific community on diet improvement to enhance health status and welfare. On this regards, at Lines 46, before the sentence “Recently, it has become widely known that the daily intake…” Authors could add “The diet composition improvement represents a key factor to enhance the animal health status and welfare (Abbate J.M. et al., Animals 2020, 10, 2303; Abbate J.M. et al., Nutrients 2021, 13, 2137).”

The section of Materials and Methods is clear for the reader and it meticulously describes the methods applied in the study.  However, Authors should check this section and correct many punctuation errors.

I suggest to indicate the body weight of animals.

Authors should indicate the inter- and intra-assay variability for ELISA test. Moreover, Authors should specify whether, for all ELISA analysis all calibrators and samples were run in duplicate and whether samples exhibited parallel displacement to the standard curve for the ELISA analysis.

Regarding statistical analysis, did the Authors apply a normality test on data in order to assess their normal distribution?

Results section as well as Discussion section is clear and well written. The findings obtained in the study were well discussed and justified with appropriate references.

The conclusion section is well written, Authors well summarize the results and the significance of the study. However, I suggest to delete the first sentence of this section as the aim of the study has already been stated. Authors could add an introductory sentence emphasizing the rationale of the study.

The figures and well as the tables are, nice, generally good and well represent the results of the study.

Authors should check and standardize the references in the list according to journal guidelines.

Reviewer 2 Report

The manuscript is mostly carefully prepared and the information that it conveyed is of importance. 

The minor weakness is the fact that the study may not represent a major advancement in knowledge, since according to the authors, the effects of B. breve MCC 1274 have been observed in mice and in mild cognitive impaired subjects. 

Misalignment of the figures and the text occurred. These may be electronically caused, but need to be resolved. An example occurs in line 207, where the text is interposed between the graph and the legend for figure 2. Misalighnment of figure legends and texts are seen also for figure 3, 4 and 5.  In addition, figures 3, 4,  5 and 6 are small and clustered; making it difficulty to appreciate their contents. 

Methods: Line 159-160 showed that the mice were anesthetized and then intracardially perfused with 4% paraformaldehyde, without perfusing with phosphate buffered saline (PBS), that is necessary to removed blood-related potential artifacts. Since the PBS step was not used it should be determined/discussed/reasoned whether the lack of that step in the histological method affects the overall outcome of the study.     
